# CLEAR : Understanding the Reasoning Capabilities of Large Language Models

## Abstract

Despite significant progress, accurately assessing the reasoning capabilities of Large Language Models (LLMs) remains both a challenging and divisive subject. Many existing benchmarks either suffer leakage, or reflect patterns in the training data, leading to ambiguous results. We present CLEAR (Conlang Logic Evaluation And Reasoning), a novel benchmark designed to test the reasoning and problem solving capabilities of LLMs in new environments. CLEAR uses Conlangs (Constructed Languages) for few-shot translation tasks, which require some linguistic knowledge to solve, but primarily the ability to make new patterns from tokens in unfamiliar contexts using logical operations. These conlangs represent a unique challenge, as while translation examples are plentiful, these conlangs each have a unique combination of rules, are self contained, and are absent in the training corpus. We present an evaluation of current frontier models over multiple metrics as a baseline for future research. We will be releasing CLEAR as a public benchmark to drive progress towards AI systems more capable of general reasoning.

## 1 Introduction

Evaluating Large Language Models (LLMs) is a challenging task. Evaluating the abilities of LLMs to reason, is both challenging and divisive. With the use of LLMs rapidly filtering into many areas of life, from code completion (Rozière et al.) to designing proteins (Li et al.), it has never been more important to evaluate these models robustly.

We feel that insight into the true reasoning capabilities of LLMs is lacking, largely due to the complexity of measuring this performance. As this is such a nascent area of study there are many complimentary measures of reasoning, which can broadly be categorised into two domains. One is to take the contents of an advanced subject and correctly draw the right conclusions to complex problems(Rein et al.). This is crucial in areas such as law, scientific analysis, and mathematics to name a few, and will be familiar to anyone who has used an LLM as an education tool.

However, while this level of accomplishment across a field is already arguably beyond that of many humans, it does not measure the ability to adapt to new information, which for people is second instinct. This is the second kind of intelligence to evaluate. How well can the model adapt to new information, leverage what is already known, and yield correct reasoning. This second intelligence is the key component required for the kinds of scientific discovery and path towards a more general system, and we argue is under evaluated in current benchmarks. Unfortunately it is largely under evaluated because it is a very challenging problem to solve, how do you find a task that challenges the LLM that is not represented in the training data? One such attempt is the Abstract Reasoning Corpus (Chollet) which provides LLMs with geometrical problems to solve, which are visually intuitive to a human but challenging to the LLM, and crucially absent from the training data. We build on this philosophy with the proposed CLEAR benchmark.

Here, in our attempt to evaluate the reasoning abilities of LLMs, we focus on one very specific task, learning a new language. We provide the model with an example of an unseen constructed language (Conlang) with translation sentences in English and some preliminary information describing how the language works, then ask for translations to example sentences. A cartoon of this setup is shown in Fig. 1. Such constructed languages have been predominantly been found in language learning enthusiast circles as toy languages but can exhibit all the features of real natural languages, while also having the advantage of not being present in the training set. The simplicity of this task will

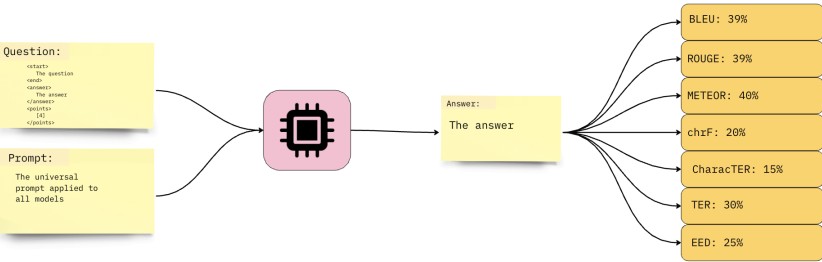

Figure 1: The translation workflow is shown here, where the question and universal prompt and provided to the LLM, and the answer evaluated with the suite of scoring mechanisms.

feel intuitive to almost everyone, when provided with only a few example translations patterns and repeated words start to become apparent, regardless or prior exposure to the language.

To 'learn' the rules of such a language is relatively simple. The translation tasks require a basic understanding of the grammatical rules and patterns with which languages are generally manipulated, and an understanding of simple logical rules. (We will revisit this concept in Sec.3.1.) Both are areas we expect modern LLMs to be proficient in considering knowledge based evaluations - this is something we aim to provide additional clarity on here. As such when the model sees an English sentence paired with a Conlang sentence we expect one sentence to be well tokenized and have good prior embeddings, and the other to be much more random. To extract a meaningful translation the model must be able to update the prior embeddings for the Conlang tokens, and make new mappings. An advantage of this task is that translation tasks are prevalent in the training data, so the pattern should be well understood, the complexity comes almost entirely from having to impose meaning onto a set of poorly initialised tokens.

We believe this poses a promising route to evaluate the in context learning capabilities of LLMs. These translation tasks show the extent to which the models can (1) use simple logical rules, to (2) update the meaning of tokens based on new information. This offers insight into the in-context learning abilities of the model, and to the reasoning ability of models to update beliefs based on new information.

**Authors Key Contributions:** In this paper we present three key contributions:

- Firstly we proposed a new dataset `CLEAR` to benchmark the ability of LLMs to reason through the medium of a translation task on new unseen languages.
- Additionally we provide initial benchmark scores for a range of current LLMs.
- Finally we also provide some analysis on the successes and limitations of current models.

## 2 RELATED WORK

**Few-Shot Prompting of Large Language Models** Vaswani et al. showed the promise of LLMs for machine translation tasks, and Brown et al. (2020) introducing GPT-3 showed that LLMs conditioned with several previously unseen examples is able to adapt to novel tasks without updating model parameters, in a method known as few-shot prompting.

Posing all tasks as sequence to sequence problems as in Chowdhery et al. (2022) and Raffel et al. (2023) makes the task of injecting few shot prompts simple to apply to a wide range of tasks, including translation. Multilingual models have been shown to exhibit the same behaviour (Lin et al., 2022) even in low resource cases, due to the ability to utilize high resource language pairs.

All current frontier models are trained on a diverse mix of natural languages (Touvron et al.Dubey et al.OpenAI et al.), which has yielded improved translation ability. Although as was found in Workshop et al. the multilingual ability is strongly correlated with data quality, particularly in low

resource languages. However, despite strong progress, translation of completely unseen languages requires either reasoning capability, or significant utilisation of other translation tasks in the training corpus.

**Logical Reasoning of Large Language Models** Recent work has shown signs that LLMs are capable of performing tasks that require logical reasoning and problem solving, despite largely being trained on next-token-prediction, many benchmarks show exceptional exam performance in fields such as law (OpenAI et al.), PhD level scientific domains (OpenAI, 2024), and mathematics (Zhang et al.). The source of this phenomena is an area of open debate, as either representative of an internal world view, or repeating patterns from the training data. Many examples claim to prove or disprove this using, for example, classic riddles and then twisting the setting subtly.

More concretely, Wei et al. proposed the *Chain of Thought* method, where the model is asked to generated additional outputs during generation akin to a stream of though, and has generally been found to improve performance on tasks requiring multi step reasoning. Kojima et al. demonstrated similar results using additional prompts like *"think step by step"*. It is argued that these results show the inherent reasoning in LLMs. Taking this approach, further work such as Zelikman et al. (2022) and Zelikman et al. (2024) starts to train a iterative loop to improve model outputs, it has been speculated that this kind of approach is behind the latest developments in OpenAI (2024). The mechanism for this in context learning is still unknown with arguments proposing effective in context gradient descent (Shen et al., 2024), however the true mechanism remains unknown.

**Logical Reasoning** There are many benchmarks quantifying aspects of the reasoning abilities of these models. Notable attempts include Srivastava et al. where a collection of tasks that require multiple steps of reasoning to complete has been organised. Other benchmarks are more specific, for example consists of questions on a short text where indirect relationships must be extracted, Liu et al. relies on a multiple choice approach, this is nice as it allows for subtle differences to be probed, and Tafjord et al. goes further requiring the model to produce from a set of facts the logical steps that justify the reasoning. Law (OpenAI et al.) and scientific (OpenAI, 2024) exam questions partly address this reasoning, but are hard to isolate from genuine facts present in the training corpus. Even strong cutting edge benchmarks such as GPQA(Rein et al.) have limitations, and are more focused on the ability to solve complex domain specific questions, which relies on specific knowledge and is subject to leakage. The different attempt by Chollet formulates geometric problems that are outside any training corpus as a hard evaluation problem.

Similarly to the step-by-step approach, many examples exist where a scratchpad is provided to the LLM to output a set of intermediate tokens in an attempt to allow the model to guide towards a local minima (Nye et al.). However, there is increasing evidence (Ye & DurrettTurpin et al.) that while the use of intermediate tokens can help reach a better output, the intermediate explanation or chain of thought is subject to just as many "hallucinations" as other outputs. Recent evaluations find the latest generation of LLMs still fail to plan multiple steps sufficiently (Valmeekam et al.).

**Translation Evaluation Metrics** Machine Translation (MT) requires automated metrics to operate at scale and to provide consistent scoring mechanisms. Traditional metrics such as BLEU (Papineni et al., 2002), METEOR (Banerjee & Lavie, 2005), ROUGE (Lin, 2004) have survived the test of time largely due to their simplicity. BLEU is arguably the most widespread as both the simplest, performing n-gram checks, and the most limited, not providing any flexibility in translation output through word stems or synonyms. As there are often many equally valid possible translations, alternative methods attempt to accommodate this. BERTScore (Zhang et al., 2020) uses a BERT (Devlin et al., 2019) model to compute similarity scores between candidate and target sentences using contextual embedding, allowing for more sophisticated translations. COMET (Rei et al., 2020) takes this approach further by using a dedicated neural network to judge the translation. However, such methods introduce their own limitations, and may perform unpredictably on languages not represented in their training corpus.

Other metrics such as chrF (Popović, 2015) and chrF++ (Popović, 2017) take the n-gram approach to characters, offering more granularity for translations missing subtle features in the output. Additionally metrics such as YiSI (Lo, 2019) attem pt to align outputs closer to human preference.

There are also edit based methods that evaluate the quality of the translation through the number of edits required to convert the candidate sentence into the reference. Metrics such as TER (Panja & Naskar, 2018) count the number of insertions, deletions, shifts, etc required to match the target

sentence. EED (Stanchev et al., 2019) extends this to better capture human preference, and CDER (Leusch et al.) and CharacTER (Wang et al., 2016) operate at the character level making them suitable to languages with rich morphology, and possibly robust to tokenization issues in LLMs.

# 3 DATASET CREATION

## 3.1 TRANSLATION EXAM SCRIPTS

The exam scripts were collated from online resources used primarily as entrance examinations for university level language degrees, however they are also popular within language learning communities online. Students applying to study Modern Foreign Languages at Universities such as Oxford University must take an entrance exam. A part of that exam is the Language Aptitude Test, which is designed to evaluate students on their ability to parse and understand a new unseen language that doesn't rely on real world knowledge.

The motivation is to standardize the test such that prior knowledge of the language doesn't provide some applicants an advantage over others, and the results should correlate with likely ability to study new languages. We propose to take this motivation and apply it to LLM evaluation.

Each exam would consist of a preamble describing the translation task, proving some details to help understand how to approach the conlang (i.e. is the language independent of word order, are the nouns gendered, etc.) then split into three or four sections. The sections then each provide a set of translation sentences, with increasing complexity as the exam progresses. Each section then has a number of questions translating from English to the Conlang, and from the Conlang to English. All translation sentences are possible to derive provided the information from the previous sections has been understood. This format maps well onto a few-shot prompt for an LLM.

## 3.2 METHODOLOGY

The dataset was constructed by scraping the publicly available exam scripts and their answers, and parsing them into individual questions as prompts for the language models. Questions for the each section all share the same preamble and translation examples, but have a different target sentence to translate. For example where section (a) might have 10 example sentences and 4 translation tasks, we repeat the stem and examples as the prompt for each of the 4 translation tasks. While there is some duplication of analysis here, this tests the ability of the model to extract the relevant patterns in different views, and the preliminary sections are required to solve the more complex later parts of the exam. As the exams are intended to be conducted as a complete exercise each question builds on knowledge derived for previous answers. Therefore for each subsequent section we append the new translation examples to the previous question stem and previous translations. This means the model has access to the example translations for (a) + (b) when answering (b), and (a) + (b) + (c) when answering (c). etc.

A simplified example of a question is given in Fig. 2. Here we highlight the patterns we might expect to extract as a human attempting to translate this task. We might start by trying to understand the word order, then find some patterns that allow us to start matching candidate word stems. Then start thinking about the endings / grammatical changes to the words based on their contexts.

## 3.3 DATASET STATISTICS AND CHARACTERISTICS

The dataset consists of 140 questions, from 12 exam scripts. There are 41 questions worth 3 points, 55 questions worth 4 points, 31 questions worth 5 points, and 13 questions worth 6 points. The maximum exam score is therefore 576 points. The complexity of the translation increases with the increase in points, and we reflect this in our '*Exam*' metrics during evaluation. Of the 140 questions 72 are ENG→CON and 68 are CON→ENG .

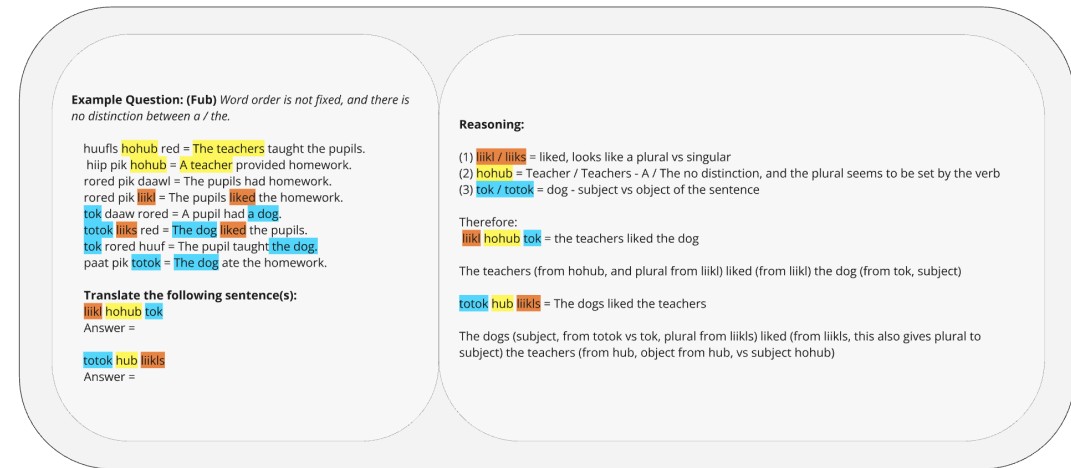

Figure 2: An example question from the dataset on the left with a (simplified) description of the language specific features. Two translation sentences are provided, but note in the real dataset only one is present at a time. On the right we show a simple way of reasoning towards the correct answer. Words are colour-coded to show which words are linked across the translation, and a more detailed description of how the translation is reached is given. This should give an indication of how a human might approach the task, we can see how it lends itself to a few-shot format iterativly building the information needed to solve the questions.

# 4 METHODOLOGY

## 4.1 MODELS EVALUATED

We have attempted to evaluate a broad a range of LLMs at a variety of model scales, ranging from the latest frontier models to small optimised LLMs. While any selection will inevitably leave some models out, this choice was made to provide representative coverage for typical use cases across a range of scales. The models evaluated fall into three categories:

**Closed Frontier Models:** These are models such as the latest GPT4o (noa), Claude (Anthropic, 2024), Gemini (Deepmind, 2024), and Mistral (Mistral, 2024). These models are broadly considered at be leading in terms of their capacities across a variety of benchmarks. These models, however, are opaque in terms of their exact architecture, prompt construction, and size. These models are all accessed via API calls.

Models: *GPT-4o, GPT-4o-Mini, GPT-3.5 Turbo, GPT-4 Turbo, Claude Opus, Claude Sonnet, Claude Sonnet 3.5, Gemini-1.5 Pro v1/2, Gemini-1.5 Flash v1/v2, Mistral Large, Mistral Small*

**Optimised Open Source Models:** We also evaluate a number of open source models. This is made possible on moderate hardware using OSS efforts such as `ollama` (Collaboration, 2024). These models are typically quantised, and efficiently reduced in size so they can be used on most modern laptops. While these optimisations will incur some small reduction in performance, it was decided this was a suitable trade off for the evaluation as this represents a common use case for LLM use in the real world.

Models: *LLama 2 70bn (Touvron et al.), Gemma 2 - 7bn, Gemma 2 2 bn  (Team et al.)*

**OpenAI: GPT-4 o1 Preview** Additionally we also studied the preview of the GPT-4-o1 (OpenAI, 2024) model from OpenAI. This model however was not available to us via the API, therefore we used the ChatGPT application to provide prompts one at a time. Each question was provided in a new chat so no memory persisted between questions. At this time, due to rate limits we have only been able to evaluate 80/140 of the questions, as such results will be provided with an asterisk until this is complete.

## 4.2 Evaluation Metrics

In our metrics we are looking for a number of properties, which are slightly different to usual translation cases. Typical translation metrics have moved away from exacting n-gram matches (Papineni et al., 2002) to allow for more flexible and natural translation (Zhang et al., 2020). While this is a significant improvement in natural language translation, we want to quantify the translation using harsh scores in order to evaluate the logical reasoning behind the translation.

The Conlang translations do not require creative solving, the vocabulary should be inferred from that used in the question stem, and the subtle variations to answers are often exactly the small grammatical errors that we are trying to evaluate the models on. To that end we have chosen the following metrics.

**BLEU:** This is our main metric, BLEU (Papineni et al., 2002) is a longstanding popular metric in machine translation due to its simplicity and interpretability. Additionally it has no dependence on the language, and the strong penalty for mistakes is preferable in the context of looking for exact translations of very tightly controlled sentences.

The BLEU metric computes $p_n$ (modified) n-gram precision scores for lengths up to $N$, weighted by $w_n$, and is given by

$$
\text{BLEU} = \begin{cases} \exp\left(\sum_{i=0}^{N} w_n \log p_n\right) & \text{if } c > r \\ \exp\left(1 - \frac{r}{c} + \sum_{i=0}^{N} w_n \log p_n\right) & \text{if } c \leq r. \end{cases} \tag{1}
$$

This score is modified by the brevity term which penalises the output sentence of length $c$ for being shorter than the target sentence of length $r$. This gives us $\text{BLEU} \in [0, 1]$. As BLEU penalises any mismatching words harshly, we enforce a strict measure on difference between words like 'horse' vs 'horses' or 'run' vs 'ran' as while the meaning may be semantically similar the error implies the underlying grammar has not been understood.

**ROUGE:** (Recall-Oriented Understudy for Gisting Evaluation) This is another n-gram matching metric, where in contrast to BLEU the main focus is recall. Recall is important in evaluating translations, as it would be possible to get a very high precision score for the produced translation, but to miss a significant fraction of the translation. BLEU takes this into account with the brevity penalty, but is an imprecise measure of how much the translation covers the target (Lin, 2004). This makes it less focused on the precision of the translation, but provides a different angle to to evaluate the outputs.

**METEOR:** The principle is to combine precision ($P$) and recall ($R$) evaluation into a single metric, uni-grams are matched between the candidate and the reference sentences and the number of matches computed (Banerjee & Lavie, 2005).

$$
\text{METEOR} = \frac{10 \cdot P \cdot R}{9 \cdot P + R}\left(1 - 0.5 \frac{\#chunks}{\#matches}\right) \tag{2}
$$

The METEOR score has a different trade off to the BLEU score, here semantic variations are allowed through the use of word stems and paraphrases. There is an obvious drawback to metrics like METEOR in this case, as the more sophisticated stages of the algorithm rely on finding matches to the stems of words, or synonyms. This is obviously not possible for Conlangs, and we will have to fall back to the basic METEOR implementation for ENG→CON translations.

**chrF:** (Character n-gram F score) follows a similar philosophy to BLEU, but computes the n-grams on the character level instead. Typically n-grams of up to 6 are chosen and the harmonic mean of precision and recall is computed. It's generally found that for morphologically rich languages, a low chrF score correlates with human preferences over metrics such as BLEU which focus on the word level **?**. The metric is computed as:

$$
\text{chrF} = (1 + \beta^2) \cdot (P \cdot R)/(\beta^2 * P + R), \tag{3}
$$

where $\beta$ controls the weighting between precision and recall. The drawbacks with this metric in common use cases are advantageous in evaluating the translations in CLEAR, the metric is sensitive to very small changes which while they may not fundamentally harm understanding, they show

where the models have not parsed the logic correctly, and the lack of semantic variation keeps the metrics clean for the Conlangs.

**EED:** (Edit Evaluation Distance) A similar concept to chrF, where the score reflects the number of edits required to transform one sentence into another. Edits include insertions, substitutions, deletions, etc. and a lower score is better (Stanchev et al., 2019).

**TER:** (Translation Edit Rate) This metric takes the same concept as EED, but looks at the word level for transformations, in a method that is designed to mirror more closely the process a human translator might take (Snover et al.).

**CharacTER:** (Character-level Translation Edit Rate) A culimation of the above metrics, which aplies the translation edit rate approach to the character level, but with awareness of the word boundaries through the addition of special characters between words (Wang et al., 2016). This results in a more sensitivity when compared to word level metrics, and better provides partial credit for translations.

**Exam Setting:** We also introduce a weighted version of some of the metrics to account for the question complexity and output sentence length. Later questions in the exams can be worth up to 6 points, compared to the 3 points per question for the early parts. Therefore we weight the output of some of the metrics (those whose outputs naturally scale between 0 and 1) by the points associated with that question, and compute a score over the entire dataset. As detailed in Sec.3.3 there is a possible total of 576 points, but for simplicity we scale this as a percentage.

### 4.3 EXPERIMENTAL SETUP

Each translation task was presented to the LLMs in a consistent way, containing the question stem, the translation examples, and the system prompt. The format of the question naturally lends itself to few-shot-prompting with the description of the task, then several lines (10 - 30) of example translations, followed by the actual question asking for a translation either from the constructed language to English or visa-via. Each translation was formatted in the same way throughout the dataset for consistency.

The system prompt was a simple but firm guide to further reinforce the 'exam' setting, and to encourage the model only to output the answer, not to re-frame the question in the response. This was designed not to enhance performance, we prefer to keep the core 'prompting' in the original question stem, and was mostly for controlling the output to encourage short answers of just the desired translation sentence.

This sort of output made it simpler to apply basic text cleaning to ensure we were evaluating the output translation and not any additional text. We decided this provided the cleanest analysis of the model's ability to reason on the raw questions without additional guidance. Several system prompts were used, and the variation in model performance as a result was deemed negligible, so the system prompt was chosen based how clean the outputs were.

## 5 RESULTS

### 5.1 OVERALL PERFORMANCE COMPARISON

The core results across the suite of benchmarks are presented in Table 1, where the best result for each metric is shown in **bold** typeface, the second best in orange, and the third best in blue. Example results are shown in Fig. 3 for two metrics, ranked by the performance, full results in Appendix.A.2.

### 5.2 MODEL RANKING

We can see clear stratification in terms of model performance, with Claude Sonnet 3.5 achieving the best results in 19/33 metrics, GPT4-o1 (preview) coming in second place with 14/33, and Claude Opus in third. The rankings are quantified in Table. 2, showing the average rank, and the number of Top-1, Top-3 and Top-5 placings for each model. This average rank performance is visualised in Fig.3b, where we can see clusters of models grouped together which have similar performance.

Table 1: Results shown for all models evaluated using all metrics. The scoring is broken down into the following categories: Evaluation method (BLEU, METEOR, etc.), translations in both directions, english to conlang, and conlang to english. These straight metrics are simple averages of the score for the translation weighting each question identically. The evaluation methods (where appropriate) have an exam setting reported, where the translation score is weighted by the points for the question, and the score over the entire question set reported. These scores are also broken into the three translation direction groups. This weighting gives more weight to more complex questions with longer prompts.

| Score | | Direction | Claude Opus | Claude Sonnet | Claude Sonnet 3.5 | GPT-3.5 Turbo | GPT-4 Turbo | GPT-4 o1 Preview* | GPT-4o | GPT-4o Mini | Llama 2 | Gemini 1.5 Flash | Gemini 1.5 Flash v2 | Gemini 1.5 Pro | Gemini 1.5 Pro v2 | Gemma 2 | Gemma 2:2b | Mistral Large | Mistral Small |
|---|---|---|---|---|---|---|---|---|---|---|---|---|---|---|---|---|---|---|---|
| BLEU | ↑ | ALL | 0.249 | 0.162 | **0.296** | 0.073 | 0.169 | 0.288 | 0.187 | 0.100 | 0.045 | 0.112 | 0.116 | 0.169 | 0.196 | 0.078 | 0.042 | 0.139 | 0.111 |
| BLEU | ↑ | CON→ENG | 0.393 | 0.263 | **0.426** | 0.106 | 0.253 | 0.417 | 0.299 | 0.159 | 0.063 | 0.195 | 0.195 | 0.265 | 0.301 | 0.121 | 0.056 | 0.218 | 0.177 |
| BLEU | ↑ | ENG→CON | 0.097 | 0.055 | 0.158 | 0.038 | 0.079 | **0.164** | 0.067 | 0.039 | 0.026 | 0.033 | 0.035 | 0.076 | 0.086 | 0.032 | 0.028 | 0.056 | 0.040 |
| BLEU Exam | ↑ | ALL | 0.243 | 0.158 | **0.282** | 0.069 | 0.153 | 0.277 | 0.175 | 0.094 | 0.043 | 0.104 | 0.113 | 0.161 | 0.188 | 0.073 | 0.041 | 0.130 | 0.107 |
| BLEU Exam | ↑ | CON→ENG | 0.404 | 0.273 | **0.426** | 0.103 | 0.236 | 0.412 | 0.290 | 0.154 | 0.062 | 0.188 | 0.197 | 0.262 | 0.299 | 0.118 | 0.055 | 0.209 | 0.177 |
| BLEU Exam | ↑ | ENG→CON | 0.092 | 0.050 | 0.147 | 0.038 | 0.075 | **0.162** | 0.066 | 0.038 | 0.024 | 0.032 | 0.036 | 0.073 | 0.084 | 0.031 | 0.027 | 0.055 | 0.040 |
| CharacTER | ↓ | ALL | 0.366 | 0.427 | **0.320** | 0.555 | 0.440 | 0.332 | 0.394 | 0.431 | 2.637 | 0.495 | 0.476 | 0.441 | 0.402 | 0.554 | 1.507 | 0.467 | 0.548 |
| CharacTER | ↓ | CON→ENG | 0.280 | 0.357 | **0.235** | 0.493 | 0.386 | 0.303 | 0.346 | 0.431 | 1.124 | 0.423 | 0.401 | 0.350 | 0.318 | 0.505 | 2.230 | 0.413 | 0.500 |
| CharacTER | ↓ | ENG→CON | 0.457 | 0.501 | 0.409 | 0.620 | 0.498 | **0.360** | 0.445 | 0.593 | 4.239 | 0.563 | 0.553 | 0.529 | 0.489 | 0.606 | 0.730 | 0.524 | 0.599 |
| chrF | ↑ | ALL | 0.571 | 0.488 | **0.620** | 0.305 | 0.483 | 0.615 | 0.509 | 0.370 | 0.212 | 0.428 | 0.416 | 0.488 | 0.520 | 0.352 | 0.228 | 0.453 | 0.376 |
| chrF | ↑ | CON→ENG | 0.678 | 0.565 | **0.711** | 0.339 | 0.531 | 0.675 | 0.581 | 0.425 | 0.259 | 0.493 | 0.480 | 0.564 | 0.600 | 0.395 | 0.250 | 0.516 | 0.458 |
| chrF | ↑ | ENG→CON | 0.458 | 0.407 | 0.524 | 0.268 | 0.433 | **0.558** | 0.432 | 0.312 | 0.163 | 0.365 | 0.350 | 0.415 | 0.437 | 0.306 | 0.204 | 0.386 | 0.289 |
| chrF Exam | ↑ | ALL | 0.568 | 0.486 | 0.614 | 0.305 | 0.475 | **0.616** | 0.499 | 0.367 | 0.206 | 0.424 | 0.417 | 0.487 | 0.520 | 0.350 | 0.227 | 0.449 | 0.374 |
| chrF Exam | ↑ | CON→ENG | 0.686 | 0.572 | 0.675 | 0.338 | 0.524 | 0.675 | 0.578 | 0.418 | 0.257 | 0.493 | 0.484 | 0.567 | 0.605 | 0.394 | 0.248 | 0.510 | 0.462 |
| chrF Exam | ↑ | ENG→CON | 0.457 | 0.405 | 0.522 | 0.273 | 0.430 | **0.565** | 0.424 | 0.320 | 0.158 | 0.366 | 0.355 | 0.418 | 0.440 | 0.308 | 0.206 | 0.391 | 0.291 |
| EED | ↓ | ALL | 0.600 | 0.695 | 0.563 | 0.840 | 0.731 | 0.582 | 0.687 | 0.795 | 3.682 | 0.812 | 0.779 | 0.729 | 0.678 | 0.842 | 1.828 | 0.739 | 0.827 |
| EED | ↓ | CON→ENG | 0.434 | 0.541 | 0.398 | 0.698 | 0.579 | 0.461 | 0.539 | 0.636 | 1.223 | 0.626 | 0.589 | 0.555 | 0.494 | 0.705 | 2.474 | 0.591 | 0.706 |
| EED | ↓ | ENG→CON | 0.777 | 0.859 | 0.738 | 0.990 | 0.893 | **0.698** | 0.845 | 0.961 | 6.286 | 0.990 | 0.975 | 0.898 | 0.871 | 0.987 | 1.135 | 0.896 | 0.955 |
| METEOR | ↑ | ALL | 0.503 | 0.413 | 0.525 | 0.264 | 0.426 | **0.564** | 0.441 | 0.319 | 0.191 | 0.336 | 0.348 | 0.419 | 0.449 | 0.291 | 0.183 | 0.394 | 0.327 |
| METEOR | ↑ | CON→ENG | 0.725 | 0.622 | 0.740 | 0.396 | 0.612 | 0.726 | 0.650 | 0.485 | 0.277 | 0.546 | 0.550 | 0.636 | 0.661 | 0.443 | 0.256 | 0.600 | 0.514 |
| METEOR | ↑ | ENG→CON | 0.268 | 0.192 | 0.298 | 0.125 | 0.228 | **0.409** | 0.219 | 0.147 | 0.099 | 0.135 | 0.141 | 0.208 | 0.228 | 0.131 | 0.105 | 0.177 | 0.129 |
| METEOR Exam | ↑ | ALL | 0.487 | 0.397 | 0.506 | 0.250 | 0.403 | **0.550** | 0.421 | 0.301 | 0.178 | 0.321 | 0.340 | 0.402 | 0.436 | 0.279 | 0.172 | 0.378 | 0.316 |
| METEOR Exam | ↑ | CON→ENG | 0.732 | 0.629 | **0.741** | 0.386 | 0.602 | 0.724 | 0.646 | 0.473 | 0.268 | 0.541 | 0.556 | 0.636 | 0.665 | 0.437 | 0.248 | 0.597 | 0.518 |
| METEOR Exam | ↑ | ENG→CON | 0.258 | 0.180 | 0.285 | 0.123 | 0.216 | **0.402** | 0.209 | 0.142 | 0.094 | 0.133 | 0.144 | 0.202 | 0.220 | 0.131 | 0.099 | 0.171 | 0.126 |
| ROUGE | ↑ | All | 0.486 | 0.394 | **0.531** | 0.258 | 0.388 | 0.521 | 0.406 | 0.305 | 0.175 | 0.314 | 0.322 | 0.381 | 0.415 | 0.277 | 0.175 | 0.352 | 0.276 |
| ROUGE | ↑ | CON→ENG | 0.660 | 0.561 | **0.695** | 0.395 | 0.533 | 0.659 | 0.551 | 0.457 | 0.262 | 0.476 | 0.489 | 0.545 | 0.591 | 0.399 | 0.240 | 0.502 | 0.397 |
| ROUGE | ↑ | ENG→CON | 0.303 | 0.217 | 0.357 | 0.113 | 0.235 | **0.390** | 0.252 | 0.147 | 0.082 | 0.158 | 0.150 | 0.222 | 0.230 | 0.147 | 0.104 | 0.192 | 0.148 |
| ROUGE Exam | ↑ | All | 0.479 | 0.385 | **0.515** | 0.251 | 0.371 | **0.515** | 0.395 | 0.296 | 0.165 | 0.304 | 0.317 | 0.373 | 0.408 | 0.272 | 0.171 | 0.339 | 0.272 |
| ROUGE Exam | ↑ | CON→ENG | 0.671 | 0.569 | **0.696** | 0.393 | 0.524 | 0.659 | 0.547 | 0.452 | 0.255 | 0.475 | 0.494 | 0.547 | 0.595 | 0.403 | 0.239 | 0.498 | 0.400 |
| ROUGE Exam | ↑ | ENG→CON | 0.299 | 0.212 | 0.346 | 0.117 | 0.227 | **0.392** | 0.253 | 0.152 | 0.082 | 0.159 | 0.155 | 0.223 | 0.232 | 0.149 | 0.107 | 0.189 | 0.152 |
| TER | ↓ | All | 0.577 | 0.672 | 0.540 | 0.818 | 0.706 | 0.547 | 0.670 | 0.777 | 3.967 | 0.797 | 0.763 | 0.703 | 0.653 | 0.828 | 1.810 | 0.712 | 0.805 |
| TER | ↓ | CON→ENG | 0.397 | 0.499 | 0.360 | 0.669 | 0.538 | 0.412 | 0.508 | 0.608 | 1.223 | 0.597 | 0.559 | 0.518 | 0.454 | 0.680 | 2.440 | 0.547 | 0.668 |
| TER | ↓ | ENG→CON | 0.767 | 0.856 | 0.730 | 0.977 | 0.885 | **0.675** | 0.842 | 0.954 | 6.873 | 0.988 | 0.972 | 0.882 | 0.862 | 0.985 | 1.132 | 0.887 | 0.950 |

Table 2: The ranking results are presented in the following table. The average position of the model across all metrics (including accounting for those where the lowest result is best). Additionally the number of times the model ranks as the Top-1, in the Top-3 or in the Top-5 results is also given. All results are also broken into the translation directions.

| score type | | Direction | Claude Opus | Claude Sonnet | Claude Sonnet 3.5 | GPT-3.5 Turbo | GPT-4 Turbo | GPT-4 o1 Preview* | GPT-4o | GPT-4o Mini | Llama 2 | Gemini 1.5 Flash | Gemini 1.5 Flash v2 | Gemini 1.5 Pro | Gemini 1.5 Pro v2 | Gemma 2 | Gemma 2:2b | Mistral Large | Mistral Small |
|---|---|---|---|---|---|---|---|---|---|---|---|---|---|---|---|---|---|---|---|
| Average Rank | ↓ | All | 3.000 | 6.818 | **1.273** | 14.818 | 7.364 | 1.727 | 4.909 | 12.364 | 16.636 | 11.091 | 10.182 | 6.818 | 4.091 | 14.000 | 16.364 | 9.000 | 12.545 |
| Average Rank | ↓ | CON→ENG | 2.273 | 6.091 | **1.000** | 14.545 | 8.000 | 2.727 | 5.364 | 12.545 | 16.000 | 10.727 | 10.091 | 6.545 | 4.000 | 14.000 | 17.000 | 9.182 | 12.909 |
| Average Rank | ↓ | ENG→CON | 3.091 | 7.545 | 2.000 | 14.182 | 5.636 | **1.000** | 5.182 | 11.545 | 17.000 | 12.091 | 11.364 | 7.182 | 4.727 | 13.727 | 16.000 | 8.636 | 12.091 |
| Top-1 Count | ↑ | All | 0 | 0 | 8 | 0 | 0 | 3 | 0 | 0 | 0 | 0 | 0 | 0 | 0 | 0 | 0 | 0 | 0 |
| Top-1 Count | ↑ | CON→ENG | 0 | 0 | 11 | 0 | 0 | 0 | 0 | 0 | 0 | 0 | 0 | 0 | 0 | 0 | 0 | 0 | 0 |
| Top-1 Count | ↑ | ENG→CON | 0 | 0 | 0 | 0 | 0 | 11 | 0 | 0 | 0 | 0 | 0 | 0 | 0 | 0 | 0 | 0 | 0 |
| Top-3 Count | ↑ | All | 11 | 0 | 11 | 0 | 0 | 11 | 0 | 0 | 0 | 0 | 0 | 0 | 0 | 0 | 0 | 0 | 0 |
| Top-3 Count | ↑ | CON→ENG | 11 | 0 | 11 | 0 | 0 | 11 | 0 | 0 | 0 | 0 | 0 | 0 | 0 | 0 | 0 | 0 | 0 |
| Top-3 Count | ↑ | ENG→CON | 10 | 0 | 11 | 0 | 0 | 11 | 1 | 0 | 0 | 0 | 0 | 0 | 0 | 0 | 0 | 0 | 0 |
| Top-5 Count | ↑ | All | 11 | 0 | 11 | 0 | 0 | 11 | 11 | 0 | 0 | 0 | 0 | 0 | 11 | 0 | 0 | 0 | 0 |
| Top-5 Count | ↑ | CON→ENG | 11 | 3 | 11 | 0 | 0 | 11 | 8 | 0 | 0 | 0 | 0 | 0 | 11 | 0 | 0 | 0 | 0 |
| Top-5 Count | ↑ | ENG→CON | 11 | 2 | 11 | 0 | 7 | 11 | 5 | 0 | 0 | 0 | 0 | 0 | 8 | 0 | 0 | 0 | 0 |

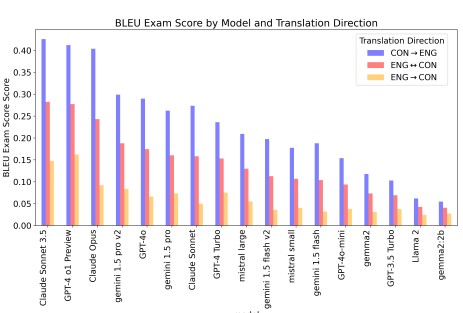 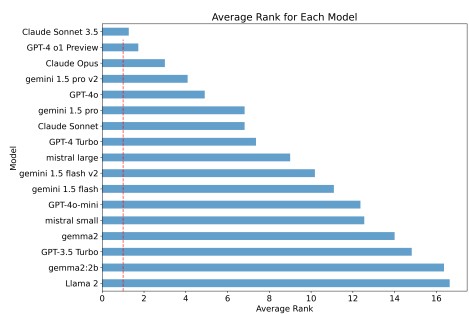

(a) Ranked results of the BLEU Exam Score for each model. Shown for translation splits.

(b) Average ranking for each model shown in order, representing the top row of Table 2.

Figure 3: Comparison of BLEU Exam Scores and model rankings across all metrics.

### 5.3 TRANSLATION DIRECTION

One striking result we find is the discrepancy between metric scores and the translation direction. When translating from CON→ENG in all cases we see the model achieve a higher scores than in the ENG→CON cases. This reduction for ENG→CON can be seen clearly in the example metrics shown in Fig.3a. This is unsurprising, as we expect models trained on a largely English language corpus to be better at generating English text, but the scale of the discrepancy hints at a deeper issue. In Fig.3a showing the results for the BLEU exam score, the ENG→CON score of top performing model (Claude Sonnet 3.5) would rank 14th when compared to the CON→ENG . We argue this is the most challenging metric in the benchmark suite, and as seen in Table.1 is dominated by GPT-4 o1.

The tasks should be identical in both directions, however the model's ability to generate grammatically correct English sentences perhaps offers an advantage, as a sentence like '*The judges hate the rams*' while unusual follows an expected English pattern. If enough information can be aggregated from the original '*kertós misóntoi hógons*' that it's something about judges (probably plural), something about hate, something about rams (again probably plural). There aren't very many ways to construct this as a grammatically correct English sentence. Additionally this is why the slightly eccentric nature of many of the translated phrases is such a desirable feature, if the model simply guesses at a sensible phrase but misses a subtle variation it will be penalised. This same phenomena is much harder to ape when doing translations of ENG→CON , and as such we find the ENG→CON BLEU exam score the strictest evaluation of the model's reasoning ability.

### 5.4 PROMPT COMPLEXITY

Slightly counter to intuition, the model performance was not strongly inversely correlated with question complexity (as measured by points). We found some models dropped in performance with more complex questions, while others increased, and some seemed to find the medium level difficulty questions the hardest before rebounding with the most complex ones. Further details of these results can be found in Appendix. A.1, but we suggest the statistical uncertainty due to limited dataset size leaves this as an area for further study.

## 6 DISCUSSION

### 6.1 WINNERS AND LOSERS

As can be seen from Table.2 the top ranks are dominated by large and recent models - suggesting the convergence of large labs on a similar capability when it comes to reasoning. However, we see the Claude family of models represented in the 1st and 3rd ranked spots, with the new GPT-4 o1 (preview) model in 2nd. The multi-step reasoning has clearly had a big impact for this latest OpenAI model compared to previous iterations, and this is seen most apparently in the ENG→CON translations where this model placed 1st in 11/11 evaluations. Despite this posed as a step change in how LLMs will be deployed in the future, the two highest placed models from Anthropic are

strongly competitive and are apparently single models. This results raises further questions on what differentiates the Anthropic dataset, or if there are simlar chain-of-thought approaches being done behind the scenes.

## 6.2 IMPLICATIONS FOR LLM CAPABILITIES

The questions in CLEAR aim to evaluate the ability of LLMs to parse new information and synthesis new meanings and connections. As the exam is based on a constructed, and therefore unseen, languages this task does not rely on memorisation, but on a basic grasp of language components, and the ability to apply fairly simply logical patterns to those components using a broad set of grammatical rules. The results are promising, especially taken for the CON→ENG direction, however the disparity found in the ENG→CON translations suggest the current state of LLMs struggle to adapt to new information and to update beliefs accordingly when the content falls outside of the training corpus. This has implications for the more open ended reasoning tasks we would ultimately wish to utilise LLMs for, and suggests significnat further work is needed.

## 6.3 LIMITATIONS OF THE STUDY

A clear limitation is the dataset size, we have a dataset consisting of 140 unique questions, from 12 different exams, so there is a fair degree of overlap between questions. For example for the same set of 10 example translations we will have 4 versions of the question stem, just with different final translation.

Another major limitation is the evaluation of the impact tokenization plays on this particular task. Conlangs by their nature often are composed of unusual strings of characters, and we suspect this plays a role in some of the difficulty the LLMs demonstrate with the task. A full evaluation of the impact of tokenization is beyond the scope of this paper.

A final major limitation of this study is linked to the power of this initial evaluation, we do not think as of submission that these questions are included in any training set we know of. Publishing this work and encouraging evaluation of the models on this dataset will no-doubt increase the availability of these example questions / answers in web-crawled datasets, removing the power of this specific study. Such dilution takes time however, and we hope this study will encourage further work in this area.

## 6.4 FUTURE WORK

Future work will require the development of a significantly larger, and more diverse set of Conlangs, however this requires a substantial effort from skilled linguists to ensure the Conlangs are both valid, and the question structure consisting of the minimally required information to derive the translation. There would also be value in exploring how this kind of in-context learning could be used to update the properties of certain tokens, and how this might lead to simple methods to circumvent safety features of current SOTA LLMs.

## 7 CONCLUSION

We have presented a new dataset benchmark, CLEAR , to evaluate the in-context reasoning capabilities of LLMs in performing translation tasks on unseen Conlangs. Our initial benchmarks across a diverse set of LLMs show promising abilities in the largest models to adapt to new information and solve complex reasoning tasks. However, we observed a significant drop in performance for tasks requiring a higher degree of reasoning, particularly in translating ENG→CON . This task, which removes the possibility of relying on the model's ability to construct grammatically correct English sentences, exposes limitations in true reasoning capabilities. These findings suggest that while progress has been made, there remains a substantial gap in general reasoning ability. We hope that by presenting as a new public benchmark, we will encourage further research targeting this specific problem. Ultimately, this work aims to pave the way towards models capable of robust, general reasoning ability across diverse and novel contexts.

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

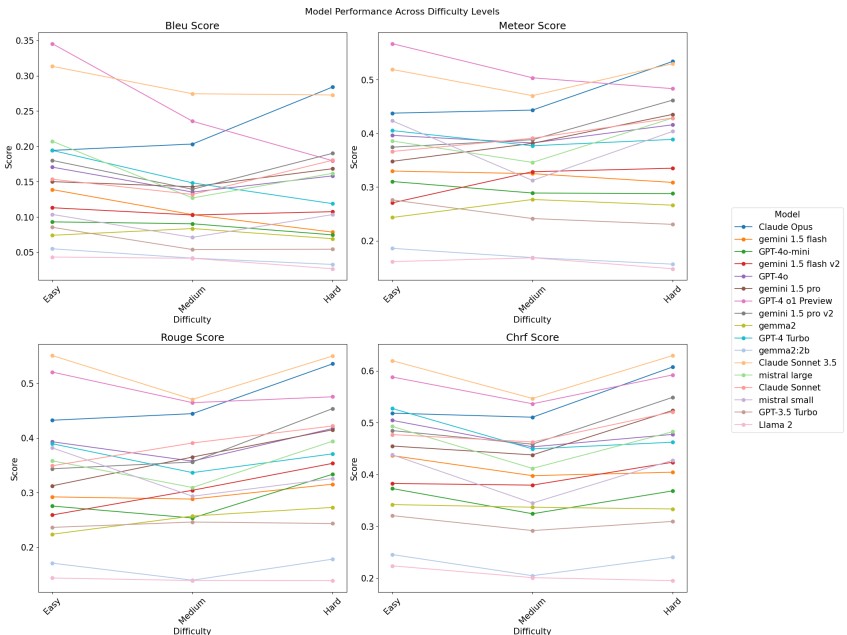

Figure 4: Plotting the relative performance for each model stratified by question difficulty - as measured by the number of points associated with the questions.

# A APPENDIX

## A.1 BY QUESTION DIFFICULTY

We investigated the performance of the models against the question difficulty, as measured by the proxy of number of points per question. The results as shown in Fig. 4 and in Table.3 there are some differences, but the results are not statistically significant considering the relatively small number of questions in each category once broken into the three difficulties. We can see however that there is not just a simple decrease in performance with complexity, the complexity is at least counteracted by the additional examples in the prompt.

| Model | BLEU Score | | | | | METEOR Score | | | | | ROUGE Score | | | | | CHRF Score | | | | |
|---|---|---|---|---|---|---|---|---|---|---|---|---|---|---|---|---|---|---|---|---|
| | Easy | Medium | Hard | %Δ | %Δ | Easy | Medium | Hard | %Δ | %Δ | Easy | Medium | Hard | %Δ | %Δ | Easy | Medium | Hard | %Δ | %Δ |
| Claude Opus | 0.194 | 0.203 | 0.284 | 4.6%, | 46.4% | 0.438 | 0.443 | 0.534 | 1.1%, | 22.0% | 0.433 | 0.445 | 0.536 | 2.8%, | 23.8% | 0.518 | 0.510 | 0.607 | -1.5%, | 17.2% |
| Claude Sonnet | 0.153 | 0.132 | 0.180 | -13.7%, | 17.6% | 0.366 | 0.391 | 0.429 | 6.8%, | 17.2% | 0.349 | 0.391 | 0.422 | 12.0%, | 20.9% | 0.477 | 0.463 | 0.521 | -2.9%, | 9.2% |
| Claude Sonnet 3.5 | 0.313 | 0.275 | 0.273 | -12.2%, | -12.8% | 0.519 | 0.470 | 0.530 | -9.4%, | 2.1% | 0.551 | 0.471 | 0.550 | -14.5%, | -0.2% | 0.619 | 0.547 | 0.629 | -11.6%, | 1.6% |
| GPT-3.5 Turbo | 0.085 | 0.054 | 0.054 | -36.6%, | -36.5% | 0.276 | 0.241 | 0.231 | -12.6%, | -16.3% | 0.236 | 0.246 | 0.244 | 4.1%, | 3.3% | 0.321 | 0.292 | 0.309 | -9.0%, | -3.7% |
| GPT-4 Turbo | 0.194 | 0.148 | 0.119 | -23.7%, | -38.8% | 0.405 | 0.377 | 0.389 | -6.9%, | -4.0% | 0.390 | 0.337 | 0.371 | -13.6%, | -4.9% | 0.527 | 0.449 | 0.462 | -14.8%, | -12.3% |
| GPT-4 o1 Preview | 0.345 | 0.236 | 0.180 | -31.7%, | -47.9% | 0.567 | 0.504 | 0.483 | -11.1%, | -14.8% | 0.521 | 0.465 | 0.475 | -10.7%, | -8.8% | 0.588 | 0.536 | 0.592 | -8.8%, | 0.7% |
| GPT-4o | 0.171 | 0.135 | 0.158 | -21.0%, | -7.5% | 0.396 | 0.382 | 0.416 | -3.5%, | 5.0% | 0.393 | 0.358 | 0.417 | -8.9%, | 6.1% | 0.504 | 0.453 | 0.477 | -10.1%, | -5.4% |
| GPT-4o-mini | 0.093 | 0.090 | 0.075 | -3.2%, | -19.4% | 0.310 | 0.289 | 0.288 | -6.8%, | -7.1% | 0.276 | 0.254 | 0.334 | -8.0%, | 21.1% | 0.373 | 0.324 | 0.368 | -13.1%, | -1.3% |
| Llama 2 | 0.043 | 0.041 | 0.027 | -4.2%, | -37.4% | 0.161 | 0.168 | 0.148 | 4.4%, | -8.1% | 0.144 | 0.139 | 0.139 | -3.3%, | -3.3% | 0.224 | 0.201 | 0.195 | -10.3%, | -12.8% |
| gemini 1.5 flash | 0.139 | 0.103 | 0.078 | -25.6%, | -43.5% | 0.330 | 0.325 | 0.309 | -1.5%, | -6.4% | 0.292 | 0.288 | 0.316 | -1.3%, | 8.2% | 0.436 | 0.398 | 0.404 | -8.7%, | -7.3% |
| gemini 1.5 flash v2 | 0.113 | 0.103 | 0.107 | -8.9%, | -5.3% | 0.271 | 0.329 | 0.335 | 21.5%, | 23.7% | 0.259 | 0.304 | 0.354 | 17.4%, | 36.6% | 0.383 | 0.379 | 0.424 | -1.0%, | 10.7% |
| gemini 1.5 pro | 0.150 | 0.143 | 0.168 | -4.7%, | 12.2% | 0.348 | 0.381 | 0.435 | 9.5%, | 25.1% | 0.313 | 0.365 | 0.415 | 16.7%, | 32.6% | 0.455 | 0.438 | 0.524 | -3.7%, | 15.2% |
| gemini 1.5 pro v2 | 0.180 | 0.139 | 0.190 | -22.7%, | 5.5% | 0.374 | 0.388 | 0.462 | 3.7%, | 23.5% | 0.344 | 0.356 | 0.454 | 3.5%, | 32.0% | 0.485 | 0.458 | 0.549 | -5.6%, | 13.2% |
| gemma2 | 0.074 | 0.084 | 0.069 | 13.5%, | -6.6% | 0.243 | 0.277 | 0.266 | 13.9%, | 9.4% | 0.224 | 0.257 | 0.273 | 14.7%, | 21.9% | 0.342 | 0.337 | 0.333 | -1.5%, | -2.6% |
| gemma2:2b | 0.055 | 0.042 | 0.033 | -23.7%, | -40.4% | 0.186 | 0.169 | 0.156 | -9.2%, | -15.9% | 0.171 | 0.140 | 0.178 | -18.1%, | 4.3% | 0.246 | 0.205 | 0.241 | -16.7%, | -2.0% |
| mistral large | 0.207 | 0.127 | 0.162 | -38.7%, | -21.7% | 0.386 | 0.346 | 0.429 | -10.3%, | 11.1% | 0.358 | 0.310 | 0.394 | -13.4%, | 10.0% | 0.493 | 0.412 | 0.483 | -16.4%, | -2.0% |
| mistral small | 0.104 | 0.071 | 0.103 | -31.7%, | -0.2% | 0.423 | 0.312 | 0.404 | -26.2%, | -4.5% | 0.382 | 0.293 | 0.326 | -23.3%, | -14.7% | 0.438 | 0.345 | 0.427 | -21.2%, | -2.5% |

Table 3: Comparison of model performance across difficulty levels Easy, Medium, Hard for various metrics. Percentages in parentheses show the relative change from Easy to Medium and Easy to Hard levels, respectively.

## A.2 ALL METRICS

We show here in Fig.5 the full set of metrics for all models split into translation directions. The same pattern can be observed across all scores with the top three models in very close competition, and with ENG→CON translations much more successfully conducted than ENG→CON .

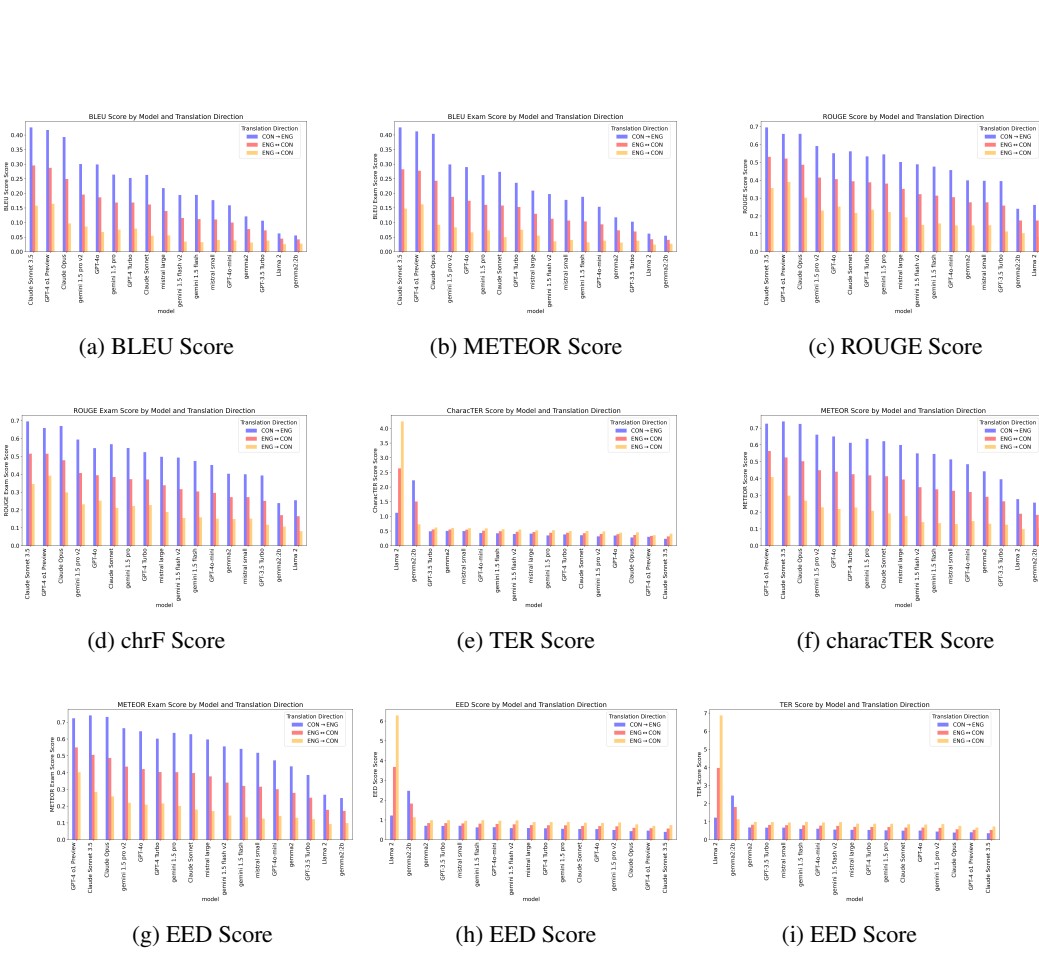

Figure 5: Comparison of Various Metric Scores by Model and Translation Direction

