# OpenReview forum: "CLEAR: Understanding the Reasoning Capabilities of Large Language Models"
_ICLR.cc/2025/Conference — ICLR 2025 Conference Withdrawn Submission_

### Official Review · Reviewer_632p · 2024-10-27

**Soundness:** 2
**Presentation:** 2
**Contribution:** 2
**Rating:** 3
**Confidence:** 3

**Summary:**

The paper introduces CLEAR, a benchmark designed specifically to assess the translation and reasoning capabilities of LLMs in novel tasks. This benchmark evaluates LLMs through few-shot translation tasks using constructed languages (conlangs)—artificial languages crafted to be unfamiliar and absent from model training data. By engaging models in translation tasks that combine logical reasoning with pattern recognition, CLEAR aims to evaluate the models’ abilities to infer grammatical rules and apply logical operations without relying on prior knowledge.

**Strengths:**

1.	This paper provides a compelling approach to evaluating LLMs on unfamiliar language tasks, offering new insights into model capabilities beyond conventional datasets.
2.	The experiment design is sufficiently thorough to support the paper’s objectives, ensuring a robust evaluation of the models' in-cotext learning capabilities.

**Weaknesses:**

1.	The "learning a new language" task feels more akin to in-context learning, emphasizing the models’ imitation and induction abilities rather than the “reasoning abilities” claimed in this paper. While "reasoning" is a term that can describe various tasks and skills, its use here in a “translation-like” task seems unsuitable.
2.	The paper devotes excessive space to fundamental concepts. For instance, the section on few-shot prompting in related work and the descriptions of common metrics like BLEU could be more concise.  Additionally, sections on logical reasoning in related work, including *Logical Reasoning of Large Language Models* and *Logical Reasoning*, are somewhat repetitive and lengthy.
3.	The paper is somewhat difficult to follow. The task remains unclear even after reading the introduction, with clarity only starting to emerge in Figure 2. Several typos are present, such as “To‘learn’” in Line 72. Additionally, the text in Figure 1 is too small to read comfortably.  Page 8 would benefit from reorganization if it only includes two tables.
4.	The analysis is somewhat superficial, focusing mainly on translation direction without offering deeper insights to inform future work. Also, could the unexpected relationship between model performance and question complexity be due to an ineffective measure of complexity?

Overall, I feel the current version seems rushed and does not meet the average ICLR standard.

**Questions:**

See weaknesses.

---

### Official Review · Reviewer_GCxR · 2024-10-31

**Soundness:** 2
**Presentation:** 2
**Contribution:** 2
**Rating:** 3
**Confidence:** 3

**Summary:**

The paper introduces CLEAR (Conlang Logic Evaluation And Reasoning), a new benchmark aimed at evaluating the reasoning capabilities of large language models (LLMs) using translation tasks between constructed languages (Conlangs) and English. Unlike natural language tasks, Conlangs present unique challenges, as they are intentionally designed to be outside the training data, reducing the risk of memorization and promoting logical reasoning. The benchmark includes translation tasks from English to Conlang and vice versa, with increasing complexity in translation rules. Popular LLMs, both closed-source and open-source, are evaluated on CLEAR, and multiple evaluation metrics, such as BLEU, ROUGE, METEOR, and character-based metrics, are applied to assess model performance on logical reasoning within language translation.

**Strengths:**

1. CLEAR is an innovative benchmark that addresses the limitations of current benchmarks by testing logical reasoning without relying on prior exposure to specific linguistic patterns. The idea of using Conlangs ensures that the task requires genuine reasoning rather than memorization or retrieval, which helps to better evaluate such ability of LLMs.
2. The paper is thorough in constructing a well-defined benchmark, with clear methodologies for dataset creation and evaluation. The variety of evaluation metrics and the stratification of translation tasks by difficulty level add rigor to the framework, enhancing the robustness of the results. Visuals such as example translation tasks (Figure 1) and the ranking results (Table 1) are useful in illustrating the key aspects of CLEAR and help readers understand the task structure.

**Weaknesses:**

1. With repeated exposure to Conlang structures, LLMs may develop specialized reasoning paths tailored to the benchmark, rather than generalizable reasoning skills. The limited dataset could falsely encourage models to adapt to specific patterns in the Conlangs, thereby reducing the benchmark’s efficacy in evaluating general reasoning capabilities.
2. The benchmark focuses on output accuracy without providing insights into the reasoning paths which is crucial for evaluating the reasoning ability. As the grammar and structure of Conlang are relatively simple, including an analysis of intermediate reasoning steps could reveal where models tend to struggle, offering diagnostic insights beyond final translation accuracy. There are related previous works like chain-of-thought, and least-to-most that prompt the model to show explicit reasoning paths. In your evaluation, do you ask the model to generate reasoning paths like in Figure 2, or it only outputs the answer? Some analysis on such paths could improve the quality of the work.
3. As mentioned by the authors, the size and task of the benchmark is quite limited. This would affect the robustness of the benchmark and could make it harder to generalize the findings.

**Questions:**

1. Are there plans to expand the dataset to include a broader range of Conlangs with varying linguistic complexities?
2. Do you have more insights on failure analysis to show if the error of existing models comes from not being able to understand the grammar, or maybe bias towards the language they're pre-trained on? For example, provide step-by-step accuracy for each question. This could also help to extend the scale of the benchmark. As mentioned in weakness 2, do you have reasoning paths generated from each LLM? What's the error rate for each step? Also, what is the distribution of different error types, e.g. mismatching between words (vocabulary), failure to identify plural (grammar), etc?
3. The identification of the translation direction's strong impact is a good direction to look into. Could the authors provide more error analysis to illustrate specific difficulties models face in this direction? Are there common patterns in errors that suggest improvements in task framing or prompting?
4. In section 4.3, you briefly mentioned the structure of the prompt including the system prompt and few-shot examples. From what I understand the examples are like the ones shown in Figure 2. It'd be better if you could include the system prompt you used in the paper as well to show how the model is guided for this specific task.

---

### Official Review · Reviewer_fw1j · 2024-11-01

**Soundness:** 3
**Presentation:** 2
**Contribution:** 3
**Rating:** 6
**Confidence:** 3

**Summary:**

This paper introduces a new reasoning benchmark CLEAR, aimed to assess the ICL and reasoning ability of LLMs.
They evaluate these abilities through introuducing new information, specifically, a new language.
Given the new information, the LLM should derive logical rules and update the meaning of tokens based on new information.
The results show that this is a challenging tasks for recent LLMs.

**Strengths:**

1. This paper studies an overlooked problem in current reasoning tasks that they, to some extent, rely on the internal knowledge of the LLMs, rather than completely on the reasoning ability.
2. CLEAR evaluates the reasoning ability in a comprehensive way, supported by staged tasks that incrementally increase in complexity. It provides a nuanced understanding of the model's capability.
3. The evaluation is comprehensive, covering a diverse range of models, and it points out the substantial room for improvement in reasoning capabilities.

**Weaknesses:**

1. The size of the dataset is limited, which is also reflected on the analysis of prompt complexity. This limitation is noted in the paper but may affect the robustness and generalizability of the findings.
2. There is a potential problem of the entanglement of tokenization and reasoning in this particular tasks. I think further analysis is required rather than just mentioning it in limitations.
3. CLEAR mainly focuses on inductive reasoning, but general reasoning often involves deductive and abductive reasoning. Improving the diversity of the task forms could significantly increase the impact.

**Questions:**

1. Is it possible to design a CON->CON task?
As you mentioned in Sec. 5.3, the knowledge of English pattern will affect the overall performance.
Since you are trying to use new information to evaluate, CON->CON is a natural thought to eliminate the influence of the English pattern.

2. Can you add the analysis of the impact of the tokenization? You also mentioned this limitation in the paper, but I assume this is an important factor to prove that CLEAR is truly evaluating the reasoning ability. I suggest creating a token-based translation that every token is not an actual word. If this setup aligns with the existing performance, readers can assume that CLEAR is truly evaluating the reasoning ability.

3. Can you add more reasoning paradigms, such as deductive reasoning, in the benchmark? I think you can provide unseens rules as premises and see whether LLMs can reason over these premises. This is an important reasoning paradigm when you want to evaluate the reasoning ability, especially when LLMs cannot do well in it.

---

### Official Review · Reviewer_ccDf · 2024-11-03

**Soundness:** 1
**Presentation:** 2
**Contribution:** 1
**Rating:** 3
**Confidence:** 4

**Summary:**

The authors propose a new task to test the reasoning capabilities of LLMs based on translation. The idea is to take standardised tests, define the rules of a new language, and ask an LLM to translate an example expressed in the new language and aided by a sufficient set of rules. By construction, the translations are not part of the training data and require simple symbol manipulation (e.g., logical operations). The authors test their benchmark on different LLMs.

**Strengths:**

The idea of using translation to test a model’s capability is interesting, especially considering that such a dataset can be scaled automatically in size and breadth. The experiments with their dataset are comprehensive and cover many sota models and standard translation metrics. Despite rushed, the article is easy to read and the ideas expressed are clear enough.

**Weaknesses:**

I have one big concern for this article.
The lack of details on what the benchmark is testing caused me some issues in understanding what capabilities it is testing. While the authors describe the task and provide a few examples, its small size (140 samples) and the lack of a detailed analysis of what kind of capabilities are required by a model to solve it (the authors mention multiple time logics and, I reckon, compositionality) make it very hard to draw comparisons with existing datasets. Furthermore, the authors do not run concurrent experiments on similar tasks (e.g., a baseline), to show correlation with existing benchmarks. In other words, is this benchmark telling us something about other popular LLMs’ benchmarks? If that’s the case, one can argue that your dataset is a proxy for some high-order reasoning and use your dataset (for example, because one can synthetically create new instances easily and does not suffer from memorisation) instead of one created by human experts.

The article seems rushed (see the paragraph on related work or the methodology, paragraph “Closed Frontier Models”).
Further, in lines 269-270, it’s your duty to run the experiments in time and before the deadline; we cannot discount the fact that GPT-4o-mini has a longer response rate, especially considering that your benchmark is very small. It’s better to present organic experiments on all the models or exclude them from the evaluation rather than preliminary results that may not be statistically significant or affected by sample bias.

Figure 1 is confusing and difficult to interpret after reading the first section. Even after reading the methodology, I still do not fully understand what it represents.

Related works are rushed, with a few references missing (line 97) or inconsistent formatting (see 107-108 vs. 122-123). Some sentences are not grammatically consistent (lines 112-13), and others express vague concepts (line 125).

**Questions:**

1) Can the authors list the advantages of using your benchmark instead of existing, human labelled or synthetic datasets for reasoning? In other words, is your benchmark a proxy of some high-order reasoning capabilities? What are the advantages of your approach over standard tests (e.g., compositionality tests that one can generalise to prevent memorisation and data leakage?).

2) Why didn’t the authors show that your results correlate (or do not) with those on popular benchmarks in reasoning and/or simple translation?

3) Say that one of those questions you use to create your benchmark is present in the training set of an LLM. How does that affect the performance of your translation? How do you ensure that a model is not interpolating the answer they potentially have to make the translation easier?

**Details Of Ethics Concerns:**

No ethical concerns

---

### Author Response · Authors · 2024-11-25
**Withdrawal**

We'd like to thank all reviewers for their time reading our paper and their constructive feedback, and are grateful for feedback that we hope will improve the core paper.
We will take on board the comments and consider how to improve the presentation of the paper and address they key research areas of concern.

And have taken the decision at this time to withdraw the paper from consideration, thank you again to the reviewers and AC.

---

### Note · Authors · 2024-11-25

**Comment:**

We'd like to thank all reviewers for their time reading our paper and their constructive feedback, and are grateful for feedback that we hope will improve the core paper. We will take on board the comments and consider how to improve the presentation of the paper and address they key research areas of concern.

And have taken the decision at this time to withdraw the paper from consideration, thank you again to the reviewers and AC.

**Withdrawal Confirmation:**

I have read and agree with the venue's withdrawal policy on behalf of myself and my co-authors.